# The predictive value of the dynamic change slope of red blood cell distribution width to platelet ratio combined with clinical indicators for the mortality outcome of patients with sepsis

Mingjuan Li, Zhonghua Lu, Lijun Cao, Yun Sun *

The First Department of Intensive Care Unit, The Second Affiliated Hospital of Anhui Medical University, Hefei, Anhui Province, China

* sunyun15@163.com

## Abstract

This retrospective study investigated the predictive value of the dynamic slope of the red blood cell distribution width to platelet ratio (RPR) for in-hospital mortality in patients with sepsis admitted to the intensive care unit (ICU). A total of 154 patients with sepsis admitted to the ICU of the Second Affiliated Hospital of Anhui Medical University between August 2023 and August 2024 were included and classified into non-survivors ($n = 37$) and survivors ($n = 117$) according to in-hospital outcome. Red blood cell distribution width (RDW) and platelet count (PLT) were recorded on days 1–5 after ICU admission, and RPR was calculated as RDW/PLT. Generalized estimating equations demonstrated significant differences in RPR between groups and over time, with a significant group-time interaction, indicating distinct temporal trends between survivors and non-survivors. The RPR in the survivors increased initially and then declined, whereas the RPR in the non-survivors showed a continuous upward trend. Receiver operating characteristic analysis showed that the slope of RPR change had good predictive performance for in-hospital mortality, with an area under the curve of 0.863 (95% CI 0.781–0.946). Optimal cutoff value of 0.017 yielded sensitivity of 86.5% and specificity of 83.8%. Multivariate logistic regression analysis identified high RPR slope (OR = 5.665, 95% CI 1.453–22.084), lactate level, Sequential Organ Failure Assessment (SOFA) score, and mechanical ventilation time as independent risk factors for in-hospital mortality. Furthermore, the combined model incorporating RPR slope, SOFA score, lactate, and mechanical ventilation time showed excellent predictive ability, with an area under the curve of 0.954 (95% CI 0.924–0.983). These findings suggest that dynamic monitoring of RPR provides valuable prognostic information and may improve early risk stratification in patients with sepsis.

**Data availability statement:** All relevant data are within the paper and its Supporting Information files.

**Funding:** Anhui Medical University Teaching Quality and Reform Project (University-Level) (Grant No. 2024xjxm91) These funding sources had no role in the study design, data collection, and analysis, or in the decision, preparation, and submission of the manuscript.

**Competing interests:** The authors have declared that no competing interests exist.

## Introduction

Sepsis is a life-threatening organ dysfunction caused by dysregulated host response to infection and remains a major global health problem with substantial mortality [1,2]. According to the Global Burden of Disease Study, sepsis was responsible for approximately 11 million deaths worldwide in 2017, accounting for nearly 20% of all global deaths [2]. At the hospital level, mortality among patients treated for sepsis remains unacceptably high. A large updated systematic review and meta-analysis reported that the pooled in-hospital mortality rate of hospital-treated sepsis was approximately 26.7% (95%CI 22.9–30.7), while mortality among patients requiring intensive care was considerably higher, with a pooled pre-discharge mortality rate of 41.9% (95% CI 36.2–47.7) [3].

Patients with severe sepsis often present with impaired immune function, compromised host defenses, multiple comorbidities, and progressive organ dysfunction, all of which contribute to poor clinical outcomes [1]. These findings highlight that sepsis continues to be associated with high mortality across different clinical settings, underscoring the need for early risk stratification and timely therapeutic intervention.

In recent years, the ratio of red blood cell distribution width to platelet count (RPR) has been shown to be associated with the prognosis of sepsis as a new predictive indicator [4-6]. Red blood cell distribution width (RDW) reflects the heterogeneity of red blood cell volume. Studies have shown that increased in RDW is associated with poor prognosis of sepsis [7], while platelet count (PLT) is positively correlated with the severity of sepsis, and decrease in PLT is an independent risk factor for death in patients with sepsis [8]. Related studies have shown that RPR is associated with adverse prognosis in patients with sepsis [4,5], severe burns [9], severe acute pancreatitis [10], etc. However, most studies only clarify the relationship between the baseline level of RPR and the prognosis of the disease, and there are few analyses on the dynamic changes of RPR and the prognosis of patients with sepsis. Therefore, this article aims to investigate the dynamic changes in RPR among patients with sepsis in the intensive care unit (ICU) and to evaluate its prognostic value.

## Materials and methods

### Research methods

This study included patients with sepsis admitted to the ICU of our hospital from August 3, 2023 to August 2, 2024. Inclusion criteria included: 1) age ≥ 18 years old 2) met the diagnostic criteria of sepsis. Sepsis was diagnosed according to the Sepsis-3 definition (increase in Sequential Organ Failure Assessment (SOFA) ≥ 2) [11] at the time of ICU admission. Exclusion criteria: 1) age < 18 years old 2) pregnant women 3) Existing end-stage or irreversible organ failure unrelated to current sepsis (e.g., advanced heart failure, end-stage renal disease requiring long-term dialysis, end-stage liver disease, or long-term ventilator dependence) 4) severe immunodeficiency, including human immunodeficiency virus infection or other conditions causing profound immune suppression 5) pre-existing hematological disorders, including hemolytic anemia (e.g., glucose-6-phosphate dehydrogenase deficiency, autoimmune

hemolytic anemia) and other major blood system diseases 6) previous thrombocytopenia or thrombocytosis 7) previous uncorrected iron deficiency anemia or megaloblastic anemia 8) any red blood cell or platelet transfusion before ICU admission or within the first five days of ICU hospitalization. Patients were divided into the non-survivors and the survivors, according to the in-hospital outcome, regardless of length of ICU stay.

Written informed consent was obtained from patients or their legally authorized representatives at ICU admission. For patients who were unconscious or unable to sign, consent was provided by family members or legal guardians. This study complies with medical ethical standards, and it was approved by the Ethics Committee of the Second Affiliated Hospital of Anhui Medical University, approval number: YX2023−131.

During the data collection process, researchers had access to patients' identifiable information in order to match electronic medical record data. However, all data were de-identified during the data analysis phase. The research protocol was approved by the ethics committee of our institution and followed the principle of patient privacy protection.

### Data collection

Clinical data collected included sex, age, body mass index (BMI), comorbidities, Acute Physiology and Chronic Health Status Evaluation II(APACHE II) score and SOFA score within 24 hours of ICU admission. Laboratory parameters included peak lactate levels during hospitalization, as well as RDW and PLT from day 1 to day 5 after ICU admission. RPR was calculated as RDW/PLT, and data were recorded for 5 days.

At our hospital, ICU patients (and those transferred to the general ward) typically undergo routine hematology testing during the early phase of critical illness. Therefore, RDW and platelet counts were generally available from day 1 to day 5 after ICU admission. Patients with missing RPR-related laboratory results within this 5-day window were excluded prior to analysis. We focused on this 5-day period to capture early dynamic changes while limiting missing data; Therefore, the final included cohort consisted only of patients with complete RPR measurements for 5 consecutive days.

In addition, we recorded other clinical information reflecting the severity of the disease, including microbial culture results, presence of septic shock, duration of mechanical ventilation, total length of hospital stay, and length of ICU stay.

### Statistical analysis

SPSS version 20.0 (IBM Corp., Armonk, NY, USA) was used for all statistical analyses. The Kolmogorov-Smirnov test was applied to assess the normality of continuous variables. Normally distributed data are presented as mean ± standard deviation and were compared between groups using the independent-samples t test. Non-normally distributed data are expressed as median (interquartile range) [M (Q1, Q3)] and were compared using the Mann-Whitney U test. Categorical variables are presented as frequencies (percentages) and were compared using the chi-square test.

All patients included in the final analysis had complete RPR measurements for five consecutive days; therefore, no imputation for missing data was performed. Generalized estimating equations (GEE) were used to model longitudinal changes in RPR and to evaluate group-by-time interactions. Pairwise comparisons of estimated marginal means (EMMs) between the non-survivors and survivors at each time point were conducted using Wald tests based on the GEE model.

To quantify individual RPR change trends, linear regression analysis was performed separately for each patient, with time as the independent variable and RPR as the dependent variable. The regression coefficient (slope) was extracted to represent the dynamic change in RPR over time. Receiver operating characteristic (ROC) curves were constructed, and the area under the curve (AUC) with 95% confidence intervals (CIs) was calculated to evaluate the predictive performance of RPR slope for in-hospital mortality. According to the optimal cutoff value derived from the Youden index, patients were classified into high and low RPR slope groups. Binary logistic regression analysis was used to identify independent risk factors for in-hospital mortality. Given the limited number of outcome events, only variables with strong clinical relevance and minimal collinearity were included in the multivariable model to reduce the risk of overfitting. A two-sided $P<0.05$ was considered statistically significant.

## Results

### Comparison of baseline clinical characteristics between the non-survivors and survivors

Compared with the survivors, the patients in the non-survivors had significant differences in ICU hospitalization time, total hospitalization time, and underlying diseases such as diabetes ($P<0.05$). However, there were no statistically significant differences in gender, age, BMI, etc. between the two groups ($P>0.05$). In terms of disease severity, septic shock, positive rate of microbial culture, mechanical ventilation time, APACHE II, and SOFA scores in the non-survivors were all higher than those in the survivors, and the difference was statistically significant ($P<0.05$). Comparison of laboratory indicators between the two groups showed that RPR from day 1 to day 5 and lactate in the non-survivors were significantly higher than those in the survivors, and the difference was significant ($P<0.001$). All analyses were performed on patients with complete day 1–5 RPR data, and no missing values were present for the longitudinal RPR time points. Baseline clinical characteristics of the non-survivors and survivors are summarized in Table 1.

### Trend and comparison of RPR between the two groups

The results of generalized estimating equation showed that the overall differences of RPR between groups (Wald$\chi^2=18.660$, df$=1$, $P<0.001$) and time points (Wald$\chi^2=17.496$, df$=4$, $P=0.002$) were statistically significant, and there was an interaction between groups and time points (Wald$\chi^2=14.966$, df$=4$, $P=0.005$). That is, the RPR of the survivors

**Table 1. Comparison of baseline clinical characteristics between the non-survivors and survivors.**

| Variables | non-survivors ($n=37$) | survivors ($n=117$) | $\chi^2$/Z/t | P value |
|---|---|---|---|---|
| Male, n (%) | 24 (64.9%) | 82 (70.1%) | 0.357 | 0.550 |
| Age (years) | 68.0 (62.0, 77.5) | 69.0 (54.5, 78.5) | −0.429 | 0.668 |
| BMI (kg/m²) | 21.5 (19.2, 24.4) | 21.5 (19.6, 24.0) | −0.489 | 0.625 |
| Comorbidities Hypertension | 15 (40.5%) | 39 (33.3%) | 0.641 | 0.423 |
| Diabetes | 7 (18.9%) | 8 (6.8%) | 4.667 | 0.031 |
| Coronary heart disease | 5 (13.5%) | 13 (11.1%) | 0.157 | 0.692 |
| Cerebrovascular disease | 4 (10.8%) | 13 (11.1%) | 0.003 | 0.959 |
| Chronic kidney disease | 2 (5.4%) | 3 (2.6%) | 0.722 | 0.395 |
| Other | 14 (37.8%) | 22 (18.8%) | 5.686 | 0.017 |
| Positive microbial culture | 29 (78.4%) | 63 (53.8%) | 7.034 | 0.008 |
| Septic shock | 37 (100%) | 73 (62.4%) | 19.480 | < 0.001 |
| Mechanical ventilation time (d) | 4.54 (2.53, 7.27) | 0.56 (0.29, 1.40) | −6.493 | < 0.001 |
| APACHE II score | 26.92±4.48 | 18.76±5.63 | 8.041 | < 0.001 |
| SOFA score | 11.0 (9.5, 12.5) | 6 (4, 8) | −7.077 | < 0.001 |
| ICU stay (d) | 8 (5, 12) | 3 (1, 5) | −5.725 | < 0.001 |
| Total length of hospital stay (d) | 12 (7, 18.5) | 14 (11, 26.5) | −2.316 | 0.021 |
| Laboratory indicators RPR (D1) | 0.12 (0.09, 0.21) | 0.07 (0.05, 0.11) | −4.259 | < 0.001 |
| RPR (D2) | 0.18 (0.12, 0.34) | 0.10 (0.07, 0.13) | −5.199 | < 0.001 |
| RPR (D3) | 0.24 (0.18, 0.60) | 0.11 (0.07, 0.15) | −6.017 | < 0.001 |
| RPR (D4) | 0.26 (0.19, 0.64) | 0.10 (0.07, 0.15) | −6.826 | < 0.001 |
| RPR (D5) | 0.30 (0.23, 1.16) | 0.08 (0.05, 0.13) | −8.089 | < 0.001 |
| Lactate (mmol·L⁻¹) | 7.5 (3.8, 11.9) | 1.9 (1.3, 3.1) | −7.490 | < 0.001 |

APACHE II: Acute Physiology and Chronic Health Status Evaluation II; SOFA: Sequential Organ Failure Assessment; RPR: Red blood Cell Distribution Width to Platelet Ratio; BMI: Body Mass Index; ICU: Intensive Care Unit. RPR was calculated as the ratio of red blood cell distribution width (RDW, %) to platelet count (PLT, ×10⁹/L). Other comorbidities included common non-life-threatening conditions such as cholelithiasis and benign prostatic hyperplasia.

and the non-survivors changed differently over time, as shown in Fig 1. Further analysis showed that the RPR of the survivors was significantly higher on the 3rd day ($P<0.001$) and 4th day ($P=0.004$) than on the 2nd day, but decreased on the 5th day, lower than the 3rd and 4th days ($P<0.001$), and also lower than the 2nd day, but not statistically significant ($P>0.05$). The RPR of the non-survivors demonstrated a progressive increase with the extension of time, and remained significantly higher on the 5th day than on the 1st day ($P<0.001$). Compared with the survivors, the RPR of the non-survivors on days 2, 3, 4, and 5 was higher than that of the survivors ($P<0.05$), especially on days 4 and 5 ($P<0.001$). On day 1, compared with the survivors, the RPR of the non-survivors demonstrated a progressive increase, but the difference was not statistically significant ($P>0.05$). See Table 2.

**Predictive value of RPR slope for in-hospital mortality in patients with sepsis**

The ROC curve of RPR slope for predicting in-hospital mortality in patients with sepsis was constructed. The area under the curve of RPR slope was 0.863 (95%CI 0.781–0.946), and the difference was statistically significant ($P<0.001$). The maximum cutoff point of Youden index was taken as the optimal cutoff value. When Youden index was 0.703, the optimal

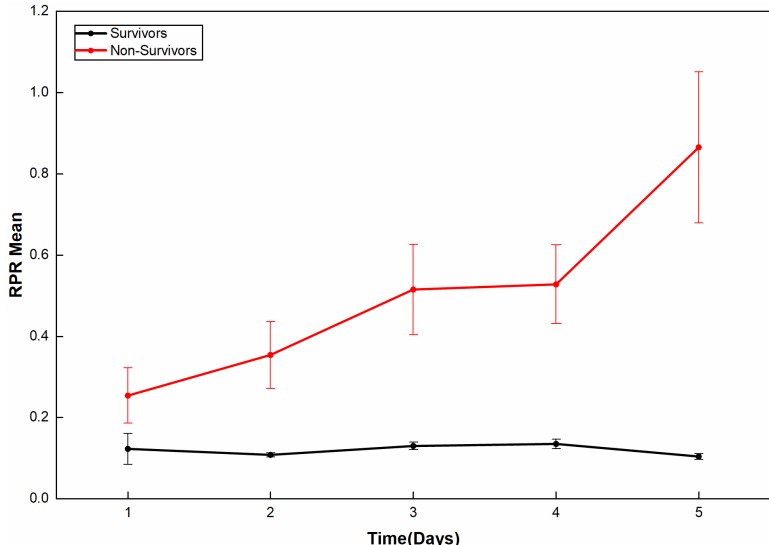

**Fig 1. Mean changes in RPR from day 1 to day 5 in the non-survivors and the survivors.** Data are presented as estimated marginal means (EMMs) ± standard error (SE).

**Table 2. Estimated marginal means (EMMs) of RPR for the survivors and the non-survivors at different time points.**

| Time | survivors (EMMs, 95% CI) | non-survivors (EMMs, 95% CI) | *P* value |
|---|---|---|---|
| Day 1 | 0.123 (0.050–0.197) | 0.254 (0.122–0.387) | 0.090 |
| Day 2 | 0.108 (0.098–0.118) | 0.354 (0.193–0.516) | 0.003 |
| Day 3 | 0.130 (0.113–0.147) | 0.515 (0.297–0.733) | 0.001 |
| Day 4 | 0.135 (0.112–0.158) | 0.528 (0.338–0.719) | < 0.001 |
| Day 5 | 0.104 (0.089–0.118) | 0.865 (0.500–1.229) | < 0.001 |

P values were obtained from Wald tests for pairwise comparisons of estimated marginal means based on the generalized estimating equation model. EMMs, estimated marginal means; RPR, red blood cell distribution width to platelet ratio; CI, confidence interval. RPR was calculated as the ratio of red blood cell distribution width (RDW, %) to platelet count (PLT, × 10⁹/L).

cutoff value for predicting in-hospital mortality in patients with sepsis was 0.017, with sensitivity of 86.5% and specificity of 83.8%. As shown in Fig 2.

## Logistic regression analysis of factors affecting the prognosis of patients with sepsis

Multivariate logistic regression analysis was performed with in-hospital mortality (non-survivors = 1, survivors = 0) as the dependent variable. Independent variables included high RPR slope (defined as RPR slope >0.017 based on ROC analysis), lactate level, SOFA score, and duration of mechanical ventilation. The results showed that high RPR slope (OR = 5.665, 95% CI 1.453–22.084), lactate (OR = 1.306, 95% CI 1.055–1.617), SOFA score (OR = 1.433, 95% CI 1.125–1.826), and mechanical ventilation time (OR = 1.168, 95% CI 1.016–1.343) were factors affecting the in-hospital mortality of patients with sepsis ($P<0.05$). See Table 3.

In this study, RPR slope, SOFA score, lactate, and mechanical ventilation time were included in the multivariate logistic regression model to establish a joint prediction model, and the ROC curve was constructed using the predicted probability. The results showed that the area under the curve of the joint model for predicting in-hospital mortality in patients with sepsis was 0.954 (95%CI 0.924–0.983), which was superior to the single factor indicators of RPR slope 0.863 (95%CI 0.781–0.946), SOFA score 0.885 (95%CI 0.820–0.949), lactate 0.909 (95%CI 0.864–0.954), and mechanical ventilation time 0.854 (95%CI 0.784–0.925), indicating strong clinical predictive performance. As shown in Fig 3.

## Discussion

In this study, in-hospital mortality was selected as the clinical outcome, as some patients experienced prolonged ICU stays exceeding 28 days, and death occurred after this time point but before hospital discharge. Using this outcome

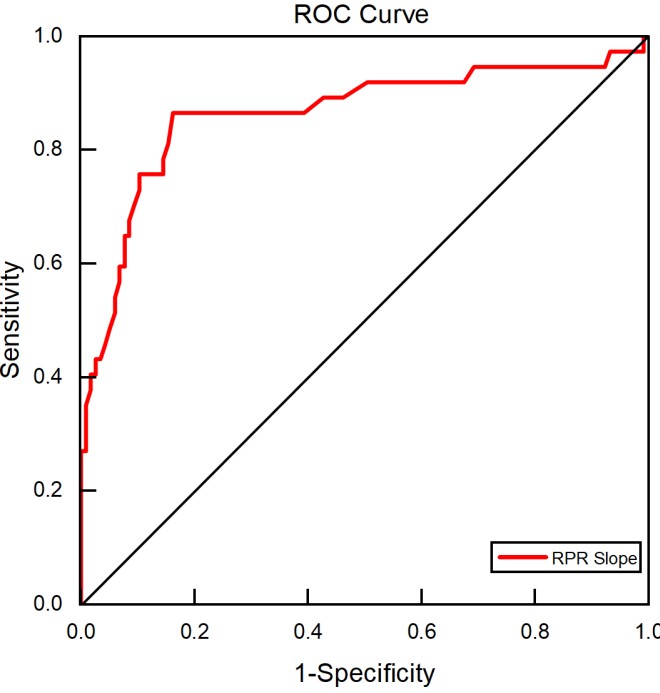

**Fig 2. ROC curve of RPR slope for predicting in-hospital mortality in patients with sepsis.** The area under the curve (AUC) was 0.863 (95% CI 0.781–0.946). RPR, red blood cell distribution width to platelet ratio; ROC, receiver operating characteristic; AUC, area under the curve.

**Table 3. Multivariate binary logistic regression analysis of risk factors for in-hospital mortality in patients with sepsis.**

| Variables | β | SE | Wald | P value | OR | 95% CI |
|---|---|---|---|---|---|---|
| High RPR slope (>0.017) | 1.734 | 0.694 | 6.241 | 0.012 | 5.665 | 1.453–22.084 |
| Lactate (mmol/L) | 0.267 | 0.109 | 6.029 | 0.014 | 1.306 | 1.055–1.617 |
| SOFA score | 0.360 | 0.124 | 8.474 | 0.004 | 1.433 | 1.125–1.826 |
| Mechanical ventilation time (d) | 0.155 | 0.071 | 4.753 | 0.029 | 1.168 | 1.016–1.343 |
| Constant | −6.925 | 1.273 | 29.609 | <0.001 | 0.001 | – |

Odds ratios (ORs) for continuous variables represent the effect per 1-unit increase. High RPR slope was analyzed as a dichotomous variable (>0.017 vs. ≤ 0.017) defined by a receiver operating characteristic (ROC)-derived cutoff.

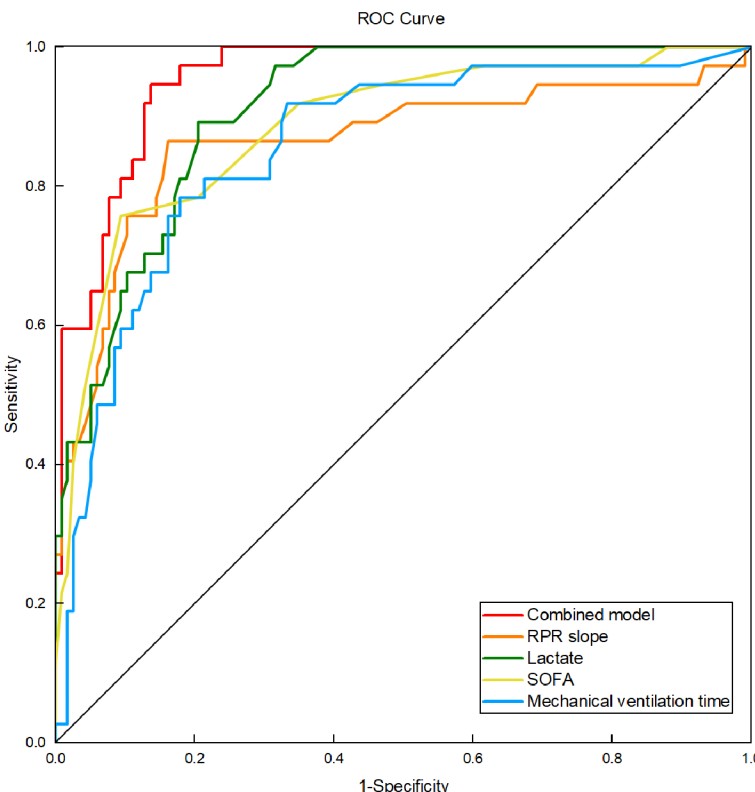

**Fig 3. ROC curve of the combined model for predicting in-hospital death in patients with sepsis.** The combined model included RPR slope, lactate level, SOFA score, and mechanical ventilation time. The ROC curves illustrate the predictive performance of each individual variable and the combined model. ROC, receiver operating characteristic; RPR, red blood cell distribution width to platelet ratio; SOFA, Sequential Organ Failure Assessment.

definition, we found that sepsis patients whose red blood cell distribution width to platelet ratio (RPR) exhibited a dynamic upward trend within the first five days after ICU admission had a significantly higher in-hospital mortality compared with those whose RPR increased slowly or decreased over time. Further logistic regression analysis demonstrated that the slope of RPR, treated as a continuous variable, independently predicted in-hospital mortality, with an area under the receiver operating characteristic curve of 0.863 (95% CI 0.781–0.946). Moreover, incorporating the RPR slope into a multivariable model together with established clinical indicators, including the SOFA score and lactate level, further improved the predictive performance.

RDW and PLT are routine detection indicators in blood routine tests, and RDW is closely related to inflammatory response [12]. However, the theory of uncontrolled inflammatory response of the body is still considered to be one of the important pathogenesis of sepsis. Studies have shown that proinflammatory factors in the body, such as tumor necrosis factor and interleukin-6, can desensitize bone marrow erythroid progenitor cells to erythropoiesis, inhibit the production and response of erythroblasts, and inhibit the maturation of erythrocytes, thereby increasing RDW [13]. In addition, cell membrane instability caused by oxidative stress, insufficient cholesterol and other substances, megaloblastic anemia caused by digestive and absorptive dysfunction caused by liver damage, renal dysfunction and other adverse factors, either alone or in combination, can increase RDW [14]. Therefore, the synergistic effects of pathological and physiological processes such as inflammatory response, oxidative stress and liver and kidney dysfunction lead to increased RDW levels in patients with sepsis and affect prognosis. Beyond its pathophysiological significance, studies have confirmed the clinical value of RDW as a prognostic biomarker for critically ill patients. In large ICU cohort studies, RDW has been shown to independently predict mortality and outperform traditional severity scores in improving risk stratification [15]. In certain critical illnesses, such as acute respiratory distress syndrome (ARDS), elevated RDW at ICU admission is associated with more severe condition and higher mortality [16]. Similarly, in patients with acute kidney injury (AKI), higher RDW levels are associated with increased short-term and long-term mortality risk [17]. These findings support the use of RDW as a convenient clinical biomarker for predicting the prognosis of critically ill patients. When sepsis occurs, inflammation in the body is unbalanced, multiple inflammatory factors increase rapidly, the coagulation system is activated, the fibrinolytic system and anticoagulation mechanism are inhibited, disseminated intravascular coagulation occurs, and a large number of platelets are consumed. Studies have confirmed that thrombocytopenia is an independent risk factor for death in patients with sepsis [8, 18]. A study reported that a simple scoring system including RDW, PLT and neutrophil index is a significant predictor of 28-day mortality in patients with severe sepsis and septic shock [19]. Therefore, RPR calculated by the ratio of RDW to PLT can reflect the state of red blood cells and platelets in the body, and thus predict the prognosis of patients with sepsis.

RPR is a newly proposed composite index with a wide range of applications, low cost and high reliability. It can comprehensively reflect the role of RDW and PLT in the progression of sepsis. Literature reports show that RPR is closely related to the prognosis of inflammatory-related diseases such as severe post-burn infection [9], acute pancreatitis [10] and rheumatoid arthritis [20]. Regarding the relationship between RPR and the prognosis of sepsis patients, related studies have shown that increased RPR levels are significantly associated with the 28-day mortality of sepsis patients [5]. Si et al. found that RPR can be combined with procalcitonin (PCT) to predict the 28-day prognosis of sepsis, and is better than a single index [21]. PRP is an independent risk factor for the 28-day prognosis of sepsis patients. The survival curve shows that patients with lower PCT and RPR have a better 28-day prognosis. Ge et al. studied adult sepsis and found that for every 0.1 increase in RPR, the risk of death at 28 days and 90 days increased by 4.0%, and the risk of ICU death increased by 6.0%. RPR is an independent risk factor for 28-day mortality, 90-day mortality, and ICU mortality in patients with sepsis [4].

However, previous studies on RPR and sepsis mostly used indicators at the time of admission to the ICU or hospital, which may not be the baseline value of RPR, and there are few studies on the dynamic evolution of RPR during sepsis. Li et al. included the changes in RPR of sepsis patients on the 1st, 2nd, 4th, and 7th days after admission to the emergency department, and found that the change trend of RPR in sepsis patients with different prognoses was different, and RPR on the 7th day of admission had good predictive value for the prognosis of sepsis [22]. Zhou et al. included patients with RPR increase of more than 30% in the first week of ICU admission in 8731 adult patients with ICU sepsis based on the MIMIC-IV database and found that sepsis patients with significantly increased RPR were positively correlated with short-term hospital mortality and length of hospital stay [6]. Continuous RPR monitoring may be an effective method to predict the short-term mortality of sepsis patients. If the inflammatory response of sepsis patients is not promptly and effectively controlled, the inflammatory response will cause continuous damage to the body, and the increased RDW and decreased

PLT will be difficult to recover, and the RPR will become higher and higher. Related studies have shown that RDW value and its changes over time are independent prognostic indicators of death in ICU sepsis patients, and combining it with SOFA, Logical Organ Dysfunction System (LODS), APACHE II and Simplified Acute Physiology Score II (SAPS II) scores can help improve the discriminative ability of these scores [23]. Studies have found that the short-term mortality of sepsis patients is only related to increased RDW, but not to baseline RDW [24]. Fawzy et al. found that the increase in RDW values at admission and on the fourth day after admission was significantly associated with mortality [25]. The increase in RDW values from day 0 to day 4 was also significantly associated with mortality. The combination of baseline RDW values and continuous RDW values can become a promising independent prognostic indicator for patients with sepsis or septic shock.

The results of this study also found that the RPR of the survivors and the non-survivors of sepsis patients changed differently over time. The RPR of the non-survivors demonstrated a progressive increase over the 5 days after admission, and was still significantly greater than that of the first day on the 5th day ($P < 0.001$). Compared with the survivors, the RPR of the non-survivors on days 2, 3, 4, and 5 was higher than that of the survivors ($P < 0.05$), especially on days 4 and 5 ($P < 0.001$). The area under the ROC curve of the RPR change slope, which reflects the 5-day RPR change trend, for predicting in-hospital mortality in patients with sepsis was 0.863, and the results of logistic regression analysis showed that high RPR slope (OR = 5.665, 95%CI 1.453–22.084) and related clinical indicators were influencing factors for in-hospital mortality in patients with sepsis ($P < 0.05$). The combined model of RPR slope, SOFA score, lactate, and mechanical ventilation time predicted in-hospital mortality in patients with sepsis with an area under the ROC curve of 0.954, which was better than any single factor indicator and consistent with the above research results.

Taken together, these findings demonstrate that RPR is not only a prognostic biomarker but also an easily accessible tool that can assist clinicians in early risk stratification and bedside dynamic monitoring. However, this study has some limitations. First, it is a single-center retrospective study with a relatively small sample size, which may introduce selection bias and limit the general applicability of the results. Second, the analysis was restricted to short-term outcomes, as only in-hospital mortality was assessed, and changes in RPR were evaluated within the first 5 days after ICU admission; therefore, the association between dynamic RPR changes and long-term prognosis could not be determined.

Although routine laboratory monitoring (including survivors transferred from ICU to general wards) typically continues beyond day 5, we limited our analysis to the first 5 days after ICU admission to minimize data gaps. Longer observation periods may provide more information, but could also lead to higher sample attrition rates and potential selection bias, especially among survivors with shorter ICU stays.

Importantly, survivors in this cohort tended to have a shorter ICU length of stay, which may reflect a lower baseline severity of illness. As a result, the observed differences in RPR dynamics between survivors and non-survivors may partially reflect underlying disease severity rather than solely representing an independent prognostic effect. Although multivariable analysis was performed to adjust for major confounders, residual confounding cannot be fully excluded. Future prospective, multicenter studies with longer follow-up are warranted to validate these findings.

Finally, the distinction between surgical and non-surgical sepsis was not consistently documented in the retrospective dataset, precluding subgroup analyses. Given that surgical status may influence both disease severity and prognosis, future prospective studies with more detailed clinical characterization are warranted.

## Conclusion

In summary, patients with different sepsis prognoses exhibit distinct continuous change trends in RPR, suggesting that dynamic changes in RPR may serve as an independent prognostic indicator for predicting mortality in patients with sepsis. Clinically, RPR should be dynamically monitored to further validate its prognostic value.

## Supporting information

**S1 Data. Original dataset of patients included in this study.**
(XLSX)

## Acknowledgments

The authors thank the clinical staff of the intensive care unit for their assistance with patient screening and data collection.

## Author contributions

**Conceptualization:** Mingjuan Li, Zhonghua Lu, Yun Sun.

**Data curation:** Mingjuan Li.

**Formal analysis:** Mingjuan Li.

**Funding acquisition:** Zhonghua Lu, Yun Sun.

**Investigation:** Mingjuan Li.

**Methodology:** Mingjuan Li.

**Resources:** Lijun Cao, Yun Sun.

**Writing – original draft:** Mingjuan Li.

**Writing – review & editing:** Mingjuan Li, Zhonghua Lu, Lijun Cao, Yun Sun.

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
