## [Decision Letter · Decision Letter 0]

5 Dec 2025

Dear Dr. Sun,

We look forward to receiving your revised manuscript.

Kind regards,

Suyan Tian

Academic Editor

PLOS ONE

**Journal Requirements:**

2. Please update your submission to use the PLOS LaTeX template. The template and more information on our requirements for LaTeX submissions can be found at http://journals.plos.org/plosone/s/latex .

4. Please upload a new copy of Figures 1, 2, and 3 as the detail is not clear. Please follow the link for more information:  https://journals.plos.org/plosone/s/figures

5. Thank you for providing your underlying data as Supporting Information.

We note that the data set contains text or data that is not in English. Please note that PLOS is an English-language publisher, so we require data sets to be provided in English as well. Please upload an English-language version of your data set.

This will also allow us to determine if your data follows PLOS standards per our Data Availability policy here: https://journals.plos.org/plosone/s/data-availability

6. Please include captions for your Supporting Information files at the end of your manuscript, and update any in-text citations to match accordingly. Please see our Supporting Information guidelines for more information: http://journals.plos.org/plosone/s/supporting-information .

**Additional Editor Comments:**

I strongly recommend that the authors consult with a statistician to review the statistical methods used and to ensure that both the analyses and their results are presented appropriately in the Methods and Results sections.

Reviewers' comments:

Reviewer's Responses to Questions

**Comments to the Author**

1. Is the manuscript technically sound, and do the data support the conclusions?

Reviewer #1: Yes

2. Has the statistical analysis been performed appropriately and rigorously?

Reviewer #1: Yes

3. Have the authors made all data underlying the findings in their manuscript fully available?

Reviewer #1: Yes

4. Is the manuscript presented in an intelligible fashion and written in standard English?

Reviewer #1: Yes

Reviewer #1: Thank you very much for the opportunity to review the manuscript entitled “The predictive value of the dynamic change slope of red blood cell distribution width to platelet ratio combined with clinical indicators for the mortality outcome of patients with sepsis”. The authors address an important topic with a well-designed hypothesis. In critically ill patients considering the dynamic changes of vital signs and laboratory measurements might be more relevant than one-time-point values for prognostication. While reviewing the article, I had the following comments and questions:

L 94 – Reference needed for number

LL 94/95 – What is “slow immunity”?

L 101 - Reference needed

L 117 – what diagnosis of sepsis was used? When was the diagnosis sepsis taken? On admission? Or generally during the ICU-stay?

LL 118 - Please elaborate further, as this contradicts the inclusion criterium of sepsis/organ dysfunction

LL 124 – How was written consent obtained from dead patients? Or was it obtained from all patients admitted with sepsis? When was it obtained? And what about sedated patients or patients incapable to provide consent. Please revise wording.

LL 168-175 – Were all patients compared or only subsets including missing data? See also example below. Survivors only stayed 3 days (mean) – how was data handled after discharge from ICU. Alternatively, this is a limitation of the analysis and conclusion

LL 177 – Table 1:

ICU stay/d: ICU stay of a mean of 3 days in the survivors group? Did blood collections occur until day 5 then? Or were these days considered as missings in the statistical analysis

Surgical and non-surgical patients with sepsis might differ in outcome. Can the authors provide numbers on the admissions of surgical and non-surgical patients. Did these groups differ in outcome?

L 194 – Please add p-values to the Table

LL 226/227 - septic shock and positive microbial probes were also significantly different between the groups, why were they not included in the Multivariate logistic regression analysis?

LL 243 ff – besides the physiology and mechanistic relevance of RDW a discussion of clinical applications would help the reader to evaluate the relevance of RDW in clinical practice. Lately, RDW has been investigated in studies on critically ill patients with e.g. ARDS, renal failure, or pancreatitis.

LL 323 ff Limitations section should be expanded a bit more. Especially survivors stayed short on the ICU. Maybe they were not sick enough?

**Do you want your identity to be public for this peer review?** For information about this choice, including consent withdrawal, please see our Privacy Policy

Reviewer #1: No

---

## [Author Response · Author response to Decision Letter 1]

6 Jan 2026

We have revised the manuscript in response to the reviewers’ and editor’s comments.

A detailed, point-by-point response is provided in the attached “Response to Reviewers” document.

---

## [Decision Letter · Decision Letter 1]

2 Feb 2026

Dear Dr. Sun,

Thank you for submitting your manuscript to PLOS ONE. After careful consideration, we feel that it has merit but does not fully meet PLOS ONE’s publication criteria as it currently stands. Therefore, we invite you to submit a revised version of the manuscript that addresses the points raised during the review process.

We look forward to receiving your revised manuscript.

Kind regards,

Suyan Tian

Academic Editor

PLOS One

Journal Requirements:

Reviewers' comments:

Reviewer's Responses to Questions

**Comments to the Author**

Reviewer #2: All comments have been addressed

Reviewer #3: All comments have been addressed

2. Is the manuscript technically sound, and do the data support the conclusions?

Reviewer #2: Yes

Reviewer #3: Yes

3. Has the statistical analysis been performed appropriately and rigorously?

Reviewer #2: Yes

Reviewer #3: Yes

4. Have the authors made all data underlying the findings in their manuscript fully available?

Reviewer #2: Yes

Reviewer #3: Yes

5. Is the manuscript presented in an intelligible fashion and written in standard English?

Reviewer #2: Yes

Reviewer #3: Yes

Reviewer #2: Dear Author,

I read the text with great interest. The subject matter is topical and your writing is excellent.

Best regards.

Reviewer #3: This manuscript has emphasized the importance of the common laboratory test including RDW and platelet ratio in the different way than the previous studies. There are a few issue needed to clarify.

- Did the author also exclude patients with hemolytic anemia e.g. G6PD deficiency or autoimmune hemolytic anemia? as those may have increased RDW as well. Please clarify.

- Did the authors exclude the patients who received RBC and platelet transfusion during day 1 to day 5 of admission? as those would affect the RDW and platelet. Please clarify.

**Do you want your identity to be public for this peer review?** For information about this choice, including consent withdrawal, please see our Privacy Policy

Reviewer #2: No

Reviewer #3: **Yes:** RUNGROTE NATESIRINILKUL

---

## [Author Response · Author response to Decision Letter 2]

5 Feb 2026

We have revised the manuscript in response to the reviewers’ and editor’s comments.

A detailed, point-by-point response is provided in the attached “Response to Reviewers” document.

---

## [Editor Report · Decision Letter 2]

10 Feb 2026

The predictive value of the dynamic change slope of red blood cell distribution width to platelet ratio combined with clinical indicators for the mortality outcome of patients with sepsis

PONE-D-25-35502R2

Dear Dr. Sun,

We’re pleased to inform you that your manuscript has been judged scientifically suitable for publication and will be formally accepted for publication once it meets all outstanding technical requirements.

Kind regards,

Suyan Tian

Academic Editor

PLOS One
---

## [Editor Report · Acceptance letter]

PONE-D-25-35502R2

PLOS One

Dear Dr. Sun,

I'm pleased to inform you that your manuscript has been deemed suitable for publication in PLOS One. Congratulations! Your manuscript is now being handed over to our production team.

Kind regards,

on behalf of

Dr. Suyan Tian

Academic Editor

PLOS One